# MAPPING THE HYPONYMY RELATION OF WORDNET ONTO VECTOR SPACES

## ABSTRACT

In this paper, we investigate mapping the hyponymy relation of WORDNET to feature vectors. We aim to model lexical knowledge in such a way that it can be used as input in generic machine-learning models, such as phrase entailment predictors. We propose two models. The first one leverages an existing mapping of words to feature vectors (*fast*Text), and attempts to classify such vectors as within or outside of each class. The second model is fully supervised, using solely WORDNET as a ground truth. It maps each concept to an interval or a disjunction thereof. On the first model, we approach, but not quite attain state of the art performance. The second model can achieve near-perfect accuracy.

## 1 INTRODUCTION

Distributional encoding of word meanings from the large corpora (Mikolov et al., 2013; 2018; Pennington et al., 2014) have been found to be useful for a number of NLP tasks. These approaches are based on a probabilistic language model by Bengio et al. (2003) of word sequences, where each word $w$ is represented as a feature vector $f(w)$ (a compact representation of a word, as a vector of floating point values).

This means that one learns word representations (vectors) and probabilities of word sequences at the same time.

While the major goal of distributional approaches is to identify distributional patterns of words and word sequences, they have even found use in tasks that require modeling more fine-grained relations between words than co-occurrence in word sequences. Folklore has it that simple manipulations of distributional word embedding vectors is inadequate for problems involving detection of other kinds of relations between words rather than their co-occurrences. In particular, distributional word embeddings are not easy to map onto ontological relations and *vice-versa*. We consider in this paper the hyponymy relation, also called the *is-a* relation, which is one of the most fundamental ontological relations.

Possible sources of ground truth for hyponymy are WORDNET Fellbaum (1998), FRAMENET Baker et al. (1998), and JEUXDEMOTS[1] (Lafourcade & Joubert, 2008). These resources have been designed to include various kinds of lexical relations between words, phrases, etc. However these resources have a fundamentally symbolic representation, which can not be readily used as input to neural NLP models. Several authors have proposed to encode hyponymy relations in feature vectors (Vilnis & McCallum, 2014; Vendrov et al., 2015; Athiwaratkun & Wilson, 2018; Nickel & Kiela, 2017). However, there does not seem to be a common consensus on the underlying properties of such encodings. In this paper, we aim to fill this gap and clearly characterize the properties that such an embedding should have. We additionally propose two baseline models approaching these properties: a simple mapping of FASTTEXT embeddings to the WORDNET hyponymy relation, and a (fully supervised) encoding of this relation in feature vectors.

---

[1]Unlike WORDNET and FRAMENET, which are developed by a teams of linguist with rigorous guidelines and strategy, JEUXDEMOTS is a resource complied by collecting the untrained users' judgments about the relationships between words, phrases etc.)

## 2    GOALS

We want to model hyponymy relation (ground truth) given by WORDNET — hereafter referred to as HYPONYMY. In this section we make this goal precise and formal. Hyponymy can in general relate common noun phrases, verb phrases or any predicative phrase, but hereafter we abstract from all this and simply write "word" for this underlying set. In this paper, we write $(\subseteq)$ for the reflexive transitive closure of the hyponymy relation (ground truth), and $(\subseteq_M)$ for relation predicted by a model $M$.[2] Ideally, we want the model to be sound and complete with respect to the ground truth. However, a machine-learned model will typically only approach those properties to a certain level, so the usual relaxations are made:

**Property 1** *(Partial soundness) A model $M$ is partially sound with precision $\alpha$ iff., for a proportion $\alpha$ of the pairs of words $w, w'$ such that $w \subseteq_M w'$ holds, $w \subseteq w'$ holds as well.*

**Property 2** *(Partial completeness) A model $M$ is partially complete with recall $\alpha$ iff., for a proportion $\alpha$ of the pairs of words $w, w'$ such that $w \subseteq w'$ holds, then $w \subseteq_M w'$ holds as well.*

These properties do not constrain the way the relation $(\subseteq_M)$ is generated from a feature space. However, a satisfying way to generate the inclusion relation is by associating a subspace of the vector space to each predicate, and leverage the inclusion from the feature space. Concretely, the mapping of words to subspaces is done by a function $P$ such that, given a word $w$ and a feature vector $x$, $P(w, x)$ indicates if the word $w$ applies to a situation described by feature vector $x$. We will refer to $P$ as a classifier. The inclusion model is then fully characterized by $P$, so we can denote it as such $((\subseteq_P))$.

**Property 3** *(space-inclusion compatibility) There exists $P : (Word \times Vec) \rightarrow [0, 1]$ such that*

$$(w' \subseteq_P w) \iff (\forall x. P(w, x) \leq P(w', x))$$

A consequence of such a model is that the relation $(\subseteq_P)$ is necessarily reflexive and transitive (because subspace inclusion is such) — the model does not have to learn this. Again, the above property will apply only to ideal situations: it needs to be relaxed in some machine-learning contexts. To this effect, we can define the measure of the subspace of situations which satisfies a predicate $p : Vec \rightarrow [0, 1]$ as follows:

$$\mathsf{measure}(p) = \int_{Vec} p(x) dx$$

(Note that this is well-defined only if $p$ is a measurable function over the measurable space of feature vectors.) We leave implicit the density of the vector space in this definition. Following this definition, a predicate $p$ is included in a predicate $q$ iff.

$$\frac{\mathsf{measure}(p \wedge q)}{\mathsf{measure}(p)} = \frac{\int_{Vec} p(x) q(x) dx}{\int_{Vec} p(x) dx} = 1$$

However, now, we can define a relaxed inclusion relation, corresponding to a proportion of $\rho$ of $p$ included in $q$:

**Property 4** *(relaxed space-inclusion compatibility) There exists $P : Word \rightarrow Vec \rightarrow [0, 1]$ and $\rho \in [0, 1]$ such that*

$$(w' \subseteq_P w) \iff \frac{\int_{Vec} P(w', x) P(w, x) dx}{\int_{Vec} P(w, x) dx} \geq \rho$$

In the following, we call $\rho$ the relaxation factor.

---

[2]We note right away that, on its own, the popular metric of cosine similarity (or indeed any metric) is incapable of modeling HYPONYMY, because it is an asymmetric relation. That is to say, we may know that the embedding of "animal" is close to that of "bird", but from that property we have no idea if we should conclude that "a bird is an animal" or rather that "an animal is a bird".

## 3 Mapping WordNet over *fast*Text

Our first model of HYPONYMY works by leveraging a general-purpose, unsupervised method of generating word vectors. We use *fast*Text Mikolov et al. (2018) as a modern representative of word-vector embeddings. Precisely, we use pre-trained word embeddings available on the *fast*Text webpage, trained on Wikipedia 2017 and the UMBC webbase corpus and the statmt.org news dataset (16B tokens). We call FTDom the set of words in these pre-trained embeddings.

A stepping stone towards modeling the inclusion relation correctly is modeling correctly each predicate individually. That is, we want to learn a separation between *fast*Text embeddings of words that belong to a given class (according to WordNet) from the words that do not. We let each word $w$ in *fast*Text represent a situation corresponding to its word embedding $f(w)$. Formally, we aim to find $P$ such that

**Property 5** $P(w, f(w')) = 1 \iff w' \subseteq w$

for every word $w$ and $w'$ found both in WordNet and in the pre-trained embeddings. If the above property is always satisfied, the model is sound and complete, and satisfies Property 3.

Because many classes have few representative elements relative to the number of dimensions of the *fast*Text embeddings, we limit ourselves to a linear model for $P$, to limit the possibility of overdoing. That is, for any word $w$, $P(w)$ is entirely determined by a bias $b(w)$ and a vector $\theta(w)$ (with 300 dimensions):

$$P(w, x) = \delta(\theta(w) \cdot x + b(w) > 0)$$

where $\delta(\text{true}) = 1$ and $\delta(\text{false}) = 0$.

We learn $\theta(w)$ and $b(w)$ by using logistic regression, independently for each WordNet word $w$. The set of all positive examples for $w$ is $\{f(w') \mid w' \in \text{FTDom}, w' \subseteq w\}$, while the set of negative examples is $\{f(w') \mid w' \in \text{FTDom}, w' \not\subseteq w\}$. We use 90% of positive examples for training (reserving 10% for testing) and we use the same amount of negatives.

We train and test for all the predicates with at least 10 positive examples. We then test Property 5. On the 10% of positive examples reserved for testing, we find that 89.4% (std. dev. 14.6 points) are identified correctly. On 1000 randomly selected negative examples, we find that 89.7% are correctly classified (std dev. 5.9 points). The result for positives may look high, but because the number of true negative cases is typically much higher than that of true positives (often by a factor of 100), this means that the recall and precision are in fact very low for this task. That is, the classifier can often identify correctly a *random* situation, but this is a relatively easy task. Consider for example the predicate for "bird". If we test random negative entities ("democracy", "paper", "hour", etc.), then we may get more than 97% accuracy. However, if we pick our samples in a direct subclass, such as (non-bird) animals, we typically get only 75% accuracy. That is to say, 25% of animals are incorrectly classified as birds.

To get a better intuition for this result, we show a Principal Component Analysis (PCA) on animals, separating bird from non-birds. It shows mixing of the two classes. This mixture can be explained by the presence of many uncommon words in the database (e.g. types of birds that are only known to ornithologists). One might argue that we should not take such words into account. But this would severely limit the number of examples: there would be few classes where logistic regression would make sense.

However, we are not ready to admit defeat yet. Indeed, we are ultimately not interested in Property 5, but rather in properties 1 and 2, which we address in the next subsection.

### 3.1 Inclusion of subspaces

A strict interpretation of Property 3 would dictate to check if the subspaces defined in the previous section are included in each other or not. However, there are several problems with this approach. To begin, hyperplanes defined by $\theta$ and $b$ will (stochastically) always intersect therefore one must take into account the actual density of the *fast*Text embeddings. One possible approximation would be that they are within a ball of certain radius around the origin. However, this assumption is incorrect: modeling the density is hard problem in itself. In fact, the density of word vectors is so low (due to

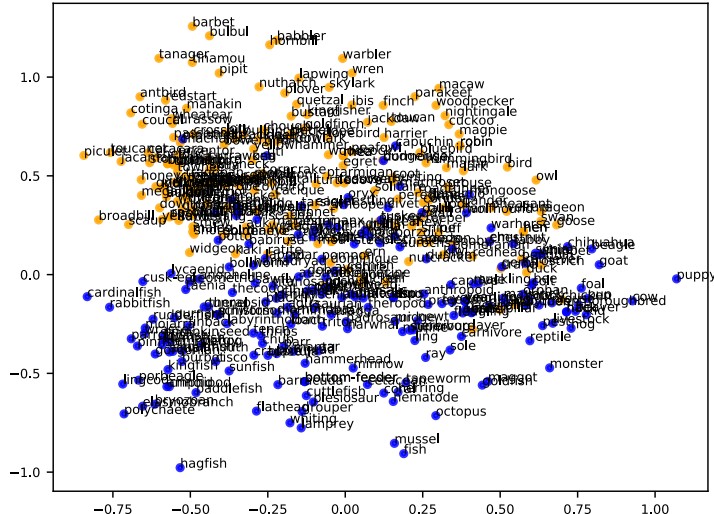

Figure 1: PCA representation of animals. Birds are highlighted in orange.

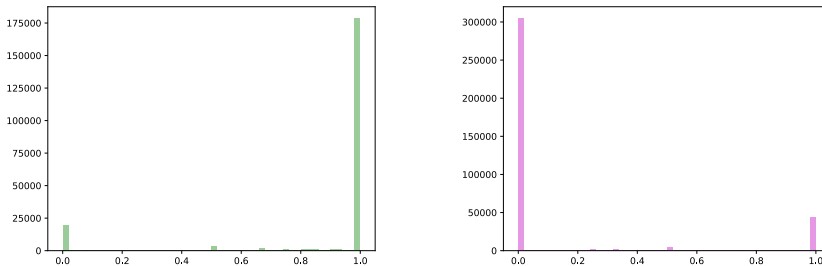

Figure 2: Results of inclusion tests. On the left-hand-side, we show the distribution of correctly identified inclusion relations in function of $\rho$. On the right-hand-side, we show the distribution of (incorrectly) identified inclusion relations in function of $\rho$.

the high dimensionality of the space) that the question may not make sense. Therefore, we refrain from making any conclusion on the inclusion relation of the euclidean subspaces, and fall back to a more experimental approach.

Thus we will test the suitability of the learned $P(w)$ by testing whether elements of its subclasses are contained in the superclass. That is, we define the following quantity

$$Q(w', w) = average\{P(w', x) \mid x \in \mathsf{FTDom}, P(w, f(x))\}$$

which is the proportion of elements of $w'$ that are found in $w$. This value corresponds to the relaxation parameter $\rho$ in Property 4.

If $w' \subseteq w$ holds, then we want $Q(w', w')$ to be close to 1, and close to 0 otherwise. We plot (figure 2) the distribution of $Q(w, w')$ for all pairs $w' \subseteq w$, and a random selection of pairs such that $w' \not\subseteq w$. The negative pairs are generated by taking all pairs $(w', w)$ such that $w' \subseteq w$, and generate two pairs $(w_1, w)$ and $(w', w_2)$, by picking $w_1$ and $w_2$ at random, such that neither of the generated pairs is in the HYPONYMY relation. We see that most of the density is concentrated at the extrema. Thus, the exact choice of $\rho$ has little influence on accuracy for the model. For $\rho = 0.5$, the recall is 88.8%. The ratio of false positives to the total number of negative test cases is 85.7%. However, we

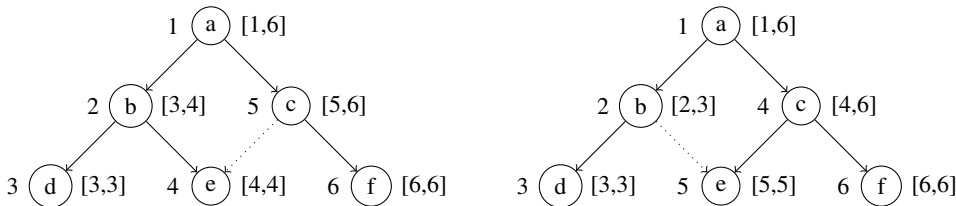

Figure 3: Two trees underlying the same dag. Notes are labeled with their depth-first index on the left and their associated interval on the right. Removed edges are drawn as a dotted line.

have a very large number of negatives cases (the square of the number of classes, about 7 billions). Because of this, we get about 1 billion false positives, and the precision is only 0.07%. Regardless, the results are comparable with state-of-the art models (section 5).

## 4 WORDNET PREDICATES AS DISJUNCTION OF INTERVALS

In this section we propose a baseline, fully supervised model model for HYPONYMY.

The key observation is that most of the HYPONYMY relation fits in a tree. Indeed, out of 82115 nouns, 7726 have no hypernym, 72967 have a single hypernym, 1422 have two hypernyms or more. In fact, by removing only 1461 direct edges, we obtain a tree. The number of edges removed in the transitive closure of the relation varies, depending on which exact edges are removed, but a typical number is 10% of the edges. In other words, when removing edges in such a way, one lowers the recall to about 90%, but the precision remains 100%. Indeed, no pair is added to the HYPONYMY relation. This tree can then be mapped to one-dimensional intervals, by assigning a position to each of the nodes, according to their index in depth-first order ($ix(w)$ below). Then, each node is assigned an interval corresponding to the minimum and the maximum position assigned to their leaves. A possible DAG and a corresponding assignment of intervals is shown in Fig. 3. The corresponding definition of predicates is the following:

$$P(w, x) = x \geq lo(w) \wedge x \leq hi(w)$$
$$lo(w) = min\{ix_T(w') \mid w' \subseteq_T w\}$$
$$hi(w) = max\{ix_T(w') \mid w' \subseteq_T w\}$$

where ($\subseteq_T$) is the reflexive-transitive closure of the $T$ tree relation (included in HYPONYMY). The pair of numbers $(lo(w), lo(w))$ fully characterizes $P(w)$. In other words, the above model is fully sound (precision=1), and has a recall of about 0.9. Additionally, Property 3 is verified.

Because it is fully sound, a model like the above one can always be combined with another model to improve its recall with no impact on precision — including itself. Such a self-combination is useful if one does another choice of removed edges. Thous, each word is characterized by an $n$-dimensional co-product (disjoint sum) of intervals.

$$w \subseteq_M w' \triangleq \bigvee_i (lo_i(w') \geq lo_i(w) \wedge hi_i(w') \leq hi_i(w))$$
$$lo_i(w) = min\{ix_{T_i}(w') \mid w' \subseteq_{T_i} w\}$$
$$hi_i(w) = max\{ix_{T_i}(w') \mid w' \subseteq_{T_i} w\}$$

By increasing $n$, one can increase the recall to obtain a near perfect model. Table 4b shows typical recall results for various values of $n$. We underline that Property 3 is not verified: the co-product of intervals do not form subspaces.

## 5 RELATED WORK

Many authors have considered modeling hyponymy. However, in many cases, this task was not the main point of their work, and we feel that the evaluation of the task has often been partially lacking.

Here, we go back to several of those and attempt to shed a new light on existing results, based on the properties presented in section 2.

## 5.1 PRECISION AND RECALL FOR HYPONYMY MODELS

Several authors, including Athiwaratkun & Wilson (2018); Vendrov et al. (2015); Vilnis et al. (2018), have proposed Feature-vector embeddings of WORDNET. Among them, several have tested their embedding on the following task: they feed their model with the transitive closure of HYPONYMY, but withhold 4000 edges. They then test how many of those edges can be recovered by their model. They also test how many of 4000 random negative edges are correctly classified. They report the average of those numbers. We reproduce here their results for this task in Table 4a. As we see it, there are two issues with this task. First, it mainly accounts for recall, mostly ignoring precision. As we have explained in section 3.1, this can be a significant problem for WORDNET, which is sparse. Second, because WORDNET is the only input, it is questionable if any edge should be withheld at all (beyond those in the transitive closure of generating edges). We believe that, in this case, the gold standard to achieve is precisely the transitive closure. Indeed, because the graph presentation of WORDNET is nearly a tree, most of the time, the effect of removing an edge will be to detach a subtree. But, without any other source of information, this subtree could in principle be re-attached to any node and still be a reasonable ontology, from a purely formal perspective. Thus we did not withhold any edge when training our second model on this task (the first one uses no edge at all). In turn, the numbers reported in Table 4a should not be taken too strictly.

| Authors, system | Result |
|---|---|
| Vendrov et al. (2015), order-embeddings | 90.6 |
| Athiwaratkun & Wilson (2018), DOE (KL) | 92.3 |
| Vilnis et al. (2018), Box + CPD | 92.3 |
| us, *fast*Text with LR and $\rho = 0.5$ | 87.2 |
| us, single interval (tree-model) | 94.5 |
| us, interval disjunctions, $n = 5$ | 99.6 |

(a) Authors, systems and respective results on the task of detection of HYPONYMY in WORDNET

| $n$ | recall |
|---|---|
| 1 | 0.91766 |
| 2 | 0.96863 |
| 5 | 0.99288 |
| 10 | 0.99973 |

(b) Typical recalls for multi-dimensional interval model. (Precision is always 1.)

Figure 4: Tables

## 5.2 PREDICATES AS SUBSPACE AND THE GEOMETRY OF THE FEATURE VECTOR SPACE

The task of modeling hyponymy is often used as an oblique way to associate subspaces to nouns (Property 3). Property 3 is an instance of the notion of order-embedding proposed by Vendrov et al. (2015), where we take subspace-inclusion as the underlying order. Vendrov et al. (2015) uses another concrete order: the intersection of (reverse) order of several scalar spaces.

A difficulty of using subspace inclusion is that, when the underlying space is sparse and high-dimensional, like section 3, it is difficult to meaningfully assign a continuous density to the feature space. One way to tackle the issue involves using lower-dimensional hyperbolic vector spaces, instead of an Euclidean one. This is proposed by (Nickel & Kiela, 2017; 2018). Nickel & Kiela (2017) makes use of the Poincaré ball model, which has been found to be useful for hosting vectorial embeddings of complex networks (Krioukov et al., 2010), and especially tree-like structures. One more model that Nickel & Kiela (2018) propose is Lorenz's hyperbolic space. Yet their aim is not to classify the feature space.

Another model was proposed by Vendrov et al. (2015), referred as a lattice model by Vilnis et al. (2018), a term which we adopt here. In this model, every vector has non-negative coordinates. The HYPONYMY relation is modeled as follows: $X$ entails $Y$ iff. for each coordinate $i$, $x_i \leq y_i$, where $(x_1, \ldots, x_n)$ and $(y_1, \ldots, y_n)$ are the $n$-dimensional vectors representing $X$ and $Y$, respectively. So, each word $w$ is associated a vector $\theta(w)$, and $P(w, x) = \delta\left(\bigwedge_i x_i \leq \theta(w)_i\right)$.

## 5.3 Equipping vector space models with probabilistic interpretations

Even though we did not list this as one of our goals, one can generalize properties 3 and 4, to go beyond inclusion, and interpret the ratio of subspace measures as a probability:

**Property 6** *(Measures correspond to probabilities)*

$$\frac{\int_{Vec} P(w', x)P(w, x)dx}{\int_{Vec} P(w, x)dx} \simeq Prob(w'|w)$$

Athiwaratkun & Wilson (2018) provide an approach to detect the HYPONYMY relation between words. They make use of (multivariate) Gaussian densities to represent words, which was first proposed by Vilnis & McCallum (2014). The idea to use Gaussian densities for representing words was also used also by Vendrov et al. (2015) and Vulic et al. (2016). However, unlike (Vilnis & McCallum, 2014; Vendrov et al., 2015; Vulic et al., 2016), but similar to what we do in our interval-based model, Athiwaratkun & Wilson advocate supervised training to build such representations for the HYPONYMY relation. Their approach outperforms those of Vendrov et al. (2015) and Vulic et al. (2016). Even though the model is based on usage of probabilistic distributions, the probabilistic nature of such representations is mainly used for modeling the uncertainty of a meaning. Plainly, they do not classify the vector space: they have no $P$ function and Property 3 is not considered. Instead, to model inclusion, which is needed for HYPONYMY detection, Athiwaratkun & Wilson (2018) make use of divergence measures, on the basis of which they define probabilistic encapsulation of two densities.

A probabilistic setting of entailment is proposed by Hockenmaier & Lai (2017), based on earlier work by Young et al. (2014); Lai & Hockenmaier (2014). The probabilistic meaning of a phrase is a random variable that encodes the ability of that phrase to describe a randomly selected image and they propose to embed phrases in a vector space (thus precisely fitting Property 6). Their phrase embeddings are inspired by the lattice model of Vendrov et al. (2015). For a phrase $X$, with the vector embedding $x = (x_1, \ldots, x_n)$, the denotational probability is defined as $p(x) = \exp(-\sum_{i=1}^{n} x_i)$. The joint probability of two phrases represented by vectors $x$ and $y$ is the probability of their join vector $z$, defined as $z_i = \max x_i, y_i$ for every $1 \leq i \leq n$. As Hockenmaier & Lai (2017) note, this approach has some imperfections. Namely, for the joint probability, it holds that:

$$p(x, y) = p(z) \geq \exp\left(-\sum_{i=1}^{n} x_i + y_i\right) = p(x)p(y)$$

Because $p(x, y) = p(x|y)p(y) = p(y|x)p(x)$, one can infer that $p(x|y) \geq p(x)$ and $p(x|y) \geq p(x)$, that is to say, that, any two phrases (words) are positively correlated.

To overcome this problem, Vilnis et al. (2018) propose instead, *box* embeddings within a unit hypercube $[0, 1]^n$, where a box stands for a hyperrectangle whose sides are parallel to coordinate axes. Each word is thus represented by a box. Vilnis et al. (2018) define then the meet and join of two boxes: their meet is an intersecting box (if any, otherwise the empty box $\perp$); the join in the minimum (in size) box that incorporates both boxes. They assume that the distribution of the underlying vector space is uniform, and so the probability of a word is defined as the volume of the corresponding box ($p(\perp)$ is set to be 0) — thus precisely fitting Property 6. As Vilnis et al. (2018) show, boxes as probabilistic random variables can have any correlation between -1 and 1, and thus avoids the aforementioned problems of the previous approach. In addition, Vilnis et al. (2018) induce probabilistic reasoning over WORDNET concepts: a node $x$ (of a graph representation of WORDNET) is assigned a probability $p(x) = N_x/N$ where $d_x$ is a number of descendant nodes of $x$ and $N$ is the total number of nodes in the graph. The joint probability of two nodes $x$ and $y$ is defined as $p(x, y) = N(x, y)/L$ where $N(x, y)$ is the total number of co-occurrences of $x$ and $y$ as the ancestors of the same leaf; and $L$ is the total number of leaves. (This is very close to the probabilities assigned by our simple 1-dimensional interval model.) Having available $p(x, y)$, $p(x)$, and $p(y)$, one computes $p(x|y)$, and $p(y|x)$. This is further used by Vilnis et al. (2018) to augment the training data with "soft edges" (if $p(x|y) > p(y|x)$ and $p(x|y)$ is sufficiently high, Vilnis et al. (2018) argue that they can be used in order to prune graphs without adding cycles). The data from WORDNET augmented in this way is used for training by Vilnis et al. (2018) to detect HYPONYMY. Their performance in terms of accuracy is the same as of Athiwaratkun & Wilson (2018)'s model (see Table 4a).

## 6  DISCUSSION, FUTURE WORK, AND CONCLUSION

We found that defining the problem of representing HYPONYMY in a feature vector is not easy. Difficulties include 1. the sparseness of data, 2. whether one wants to base inclusion on an underlying (possibly relaxed) inclusion in the space of vectors, and 3. determining what one should generalize.

Our investigation of WORDNET over *fast*Text demonstrates that WORDNET classes are not cleanly linearly separated in *fast*Text, but they are sufficiently well separated to give a useful recall for an approximate inclusion property. Despite this, and because the negative cases vastly outnumber the positive cases, the rate of false negatives is still too high to give any reasonable precision. One could try to use more complex models, but the sparsity of the data would make such models extremely sensitive to overfitting.

Our second model takes a wholly different approach: we construct intervals directly from the HYPONYMY relation. The main advantage of this method is its simplicity and high-accuracy. Even with a single dimension it rivals other models. A possible disadvantage is that the multi-dimensional version of this model requires disjunctions to be performed. Such operations are not necessarily available in models which need to make use of the HYPONYMY relation. At this stage, we make no attempt to match the size of intervals to the probability of a word. We aim to address this issue in future work.

Finally, one could see our study as a criticism of WORDNET as a natural representative of HYPONYMY. Because it is almost structured like a tree, one can suspect that it in fact misses many hyponymy relations. This would also explain why our simple *fast*Text-based model predicts more relations than present in WORDNET. One could think of using other resources, such as JEUXDEMOTS. Our preliminary investigations suggest that these seem to suffer from similar flaws — we leave complete analysis to further work.

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
