# OpenReview forum: "Mapping the hyponymy relation of wordnet onto vector Spaces"
_ICLR.cc/2019/Conference_

### Official Review · AnonReviewer2 · 2018-11-01
**An interesting perspective on modeling hyponymy, but doesn't make it over the bar**

**Rating:** 3
**Confidence:** 5

**Review:**

This paper reads like the first thoughts and experiments of a physicist or mathematician who has decided to look at word representations and hyponymy. I mean that in both the positive and negative ways that this remark could be construed. On the positive side, it provides an interesting read, with a fresh perspective, willing to poke holes in rather than accepting setups that several previous researchers have used.  On the negative side, though, this paper can have an undue, work-from-scratch mathiness that doesn't really contribute insight or understanding, and the current state of the work is too preliminary. I think another researcher interested in this area could benefit from reading this paper and hearing the perspective it presents. Nevertheless, there just isn't sufficient in the way of strong, non-trivial results in the current paper to justify conference acceptance.

Quality:

 - Pro
   o Everything is presented in a precise formalized fashion. The paper has interesting remarks and perspectives. I appreciate that the authors not only did find most existing work on modeling hyponymy but provide a detailed and quite insightful discussion of it.  (A related paper from overlapping authors to papers you do cite that maybe should have been included is Chang et al. https://arxiv.org/abs/1710.00880 – which is a bit different in trying to learn hyponyms from text not WordNet, but still clearly related.)
  -Con
   o There just isn't enough here in the way of theoretical or experimental results. In the end, two "methods" of hyponymy modeling are presented: one is a simple logistic regression, which is estimated separately PER WORD for words with 10 or more hyponyms. This performs worse than the methods of several recent papers that the author cites. The other is a construction that shows that any tree can be embedded by representing nodes as ranges of the real line. This is true, but trivial. Why don't ML/NLP researchers do this? It's because they want a representation that doesn't only represent the ISA hierarchy but also other aspects of word meaning such as meaning similarity and dimensions of relatedness. Furthermore, in general they would like to learn these representations from data rather than hand-constructing it from an existing source like WordNet. For instance, simply doing that gives no clear way to add other words not in wordnet into the taxonomy. This representation mapping doesn't really give any clear advantage beyond just looking up hyponymy relationships in wordnet when you need them.

Clarity:
 - Pro
   o The paper is in most respects clearly written and enjoyable to read.
 - Con
   o The mathematical style and precision has it's uses, but sometime it just seemed to make things harder to follow. Referring to things throughout as "Property k" – even though some of those properties were given names when first introduced – left me repeatedly flicking up and down through the PDF to refresh myself on what claim was being referred to without any apparent need....

Originality:
 - Pro
   o There is certainly originality of perspective. The authors make some cogent observations on how other prior work has been naive about adopted assumptions and as to what it has achieved (e.g., in the discussion at the start of section 5.1).
 - Con
   o There is not really significant originality of method. The logistic regression model is nothing but straightforward. (It is also highly problematic in learning a separate model for each word with a bunch of hyponyms. This both doesn't give a model that would generalize to novel words or ones with few hyponyms.) Mapping a tree to an interval is fairly trivial, and besides this is just a mapping of representations, it isn't learning a good representation as ML people (or ICLR people) would like. The idea that you can improve recall by using a co-product (disjunction) of intervals is cute, though, I admit. Nice.

Significance
 - Con
   o I think this work would clearly need more development, and more cognizance of the goals of generalizable representation learning before it would make a significant contribution to the literature.

Other:
 - p.1: Saying about WordNet etc., "these resources have a fundamentally symbolic representation, which can not be readily used as input to neural NLP models" seems misplaced when there is now a lot of work on producing neural graph embeddings (including node2vec, skip-graphs, deepwalk, etc.). Fundamentally, it is just a bad argument: It is no different to saying that words have a fundamentally symbolic representation which cannot be readily used as input to neural NLP models, but the premise of the whole paper is already that we know how to do that and it isn't hard through the use of word embeddings.
 - p.2: The idea of words and phrases living in subset (and disjointness etc.) relationships according to denotation is the central idea of Natural Logic approaches, and these might be cited here. There are various works, some more philosophical. A good place to start might be: https://nlp.stanford.edu/pubs/natlog-iwcs09.pdf
 - p.2: The notions of Property 1 and 2 are just "precision" and "recall", terms the paper also uses. Do we gain from introducing the names "Property 1" and "Property 2" for them? I also felt that I wouldn't have lost anything if Property 3 was just the idea that hyponymy is represented as vector subspace inclusion.
 - p.2: fn.2: True, but it seems fair to more note that cosine similarity is very standard as a word/document similarity measure, not for modeling hyponymy, for this reason.
 - p.4: Below the equation, shouldn't it be Q(w', w) [not both w'] and then Q(w', w) and not the reverse? If not, I'm misunderstanding.

---

### Official Review · AnonReviewer1 · 2018-11-03
**Interesting problem, but contribution (and clarity) currently inadequate**

**Rating:** 3
**Confidence:** 3

**Review:**

This paper explores the notion of hyponymy in word vector representations. It tests the capacity of a logistic regression classifier to distinguish words that are and are not hyponyms using fastText embeddings, and it also describes a method of organizing WordNet relations into a tree structure and defining hyponymy based on this structure.

The problem of capturing hyponymy relations within vector space models of word representation is an interesting and important one, but it is not clear to me that this paper has made a substantive contribution to it. The paper seems simply to 1) observe that fastText embeddings are imperfect for hyponymy detection with a linear classifier, and 2) reconstruct the fairly natural interpretation of WordNet relations as a hierarchical tree structure, and to re-extract hyponymy relations from that tree structure. As far as I can tell, the paper's “supervised” model does not use embeddings  (or learning) at all.

Assessing the paper's contribution is made more difficult by an overall lack of clarity. The details of the experiments are not laid out with sufficient explicitness, and the reporting of results is also fairly confusing (I am not clear, for example, on what is depicted in Figure 2). The paper is not organized in a particularly intuitive way, nor has it made clear what the contributions might be.

Overall, while I think that this is a worthy topic, I do not think that the contribution or the clarity of this paper are currently sufficient for publication.

Additional comments:

-The PCA plot is too dense to be a useful visual - it would be more useful to plot a smaller number of relevant points.

-Results should be presented more clearly in table form - there seem to be a large number of results that are not reported in any table (for instance, the results described in Section 3).

---

### Official Review · AnonReviewer3 · 2018-11-05
**An interesting problem, but unclear description and methodological issues**

**Rating:** 3
**Confidence:** 4

**Review:**

* Summary of the paper

This paper studies how hyponymy between words can be mapped to feature representations. To this end, it lists out properties of such mappings and studies two methods from the perspective of how they address these properties.

* Review
The goal of this paper: namely, formalizing the hypernymy relation over vector spaces is not only an interesting one, but also an important one -- being able to do so can help us understand and improve vector representations of words and reason about their quality.

In its execution, however, the paper does not seem ready for publication at this point. Two major issues stand out.

First, several things are unclear about the paper. Here's a partial list:
1. Property 3 is presented as a given. But why is this property necessary or sufficient for defining hyponymy?
2. It is not clear why the measure based definition is introduced. Furthermore, the expression above the statement of property 4 is stated as following from the definition. It may be worth stating why.
3. Section 3.1 is entirely unclear. The plots in Fig 2 are empty. And in the definition of Q on page 4, the predicate that defines the set states P(w, f(x)). But if the range of P is [0, 1], what does it mean as a predicate? Does this mean we restrict it to cases where P(w, f(x)) = 1?

Second, there are methodological concerns about the experiments.
1. In essence, section 3 proposes to create a word-specific linear classifier that decides whether a new vector is a hypernym or not. But this classifier faces huge class imbalance issues, which suggests that simply training a classifier as described can not work (as the authors discovered). So it is not clear what we learn from this section? Especially because the paper says at the just before section 3.1 that "we are ultimately not interested in property 5".
2. Perhaps most importantly, the method in section 4 basically represents a pruned version of WordNet as a collection of intervals. It is not surprising that this gets high recall because the data is explicitly stored in the form of intervals. Unfortunately, however, this means that there is no generalization and the proposed representation for hyponymy essentially remembers WordNet. If we are allowed to do that, then why not just define f(w) to be an indicator for w and P(w, f(w')) to be an indicator for whether the word w is a hyponym of w'. This would give us perfect precision and recall, at the cost of no generalization.

* Minor points
1. The properties 1 and 2 are essentially saying that the precision and recall respectively are alpha. Is this correct?
2. Should we interpret P as a probability? The paper doesn't explicitly say so, but why not?
3. The paper is written in a somewhat informal style. Some examples:
   - Before introducing property 3, the paper says that it is a "satisfying way". Why/for whom?
   - The part about not admitting defeat (just above section 3.1)
   While these are not bad by themselves, the style tends to be distracting from the point of the paper.

* Missing reference
See: Faruqui, Manaal, Jesse Dodge, Sujay Kumar Jauhar, Chris Dyer, Eduard Hovy, and Noah A. Smith. "Retrofitting Word Vectors to Semantic Lexicons." In Proceedings of the 2015 Conference of the North American Chapter of the Association for Computational Linguistics: Human Language Technologies, pp. 1606-1615. 2015.

This paper and its followup work discusses relationships between word embeddings and relations defined by semantic lexicons, including hyponymy.

---

### Author Response · Authors · 2018-11-07
**General response to reviews**

Thank you for your reviews. As we see it, the reviewers missed the main point of the paper, hoping that it makes our intent clear.

We consider the task of mapping wordnet to vector spaces, giving two baselines.

1. The first baseline is based on dividing fasttext into subspaces corresponding to predicates in wordnet
2. The second baseline computes a simple embedding of predicates to intervals.

The second baseline beats the state of the art, with a much simpler method. Some reviewers claim that it does not generalise --- but we counter that no existing method generalises either (see the paper for an argumentation). So we maintain that we improve on the state of the art.

The first baseline allows for generalisation. Admittedly, its precision is low, but we point out that the same shortcoming is found in state of the art methods.

---

### Meta-Review · Area_Chair1 · 2018-12-13
**Interesting exploration of the topic, but no clear contributions**

**Confidence:** 5
**Recommendation:** Reject

**Metareview:**

All three reviewers found this to be an interesting exploration of a reasonable topic—the use of ontologies in word representations—but all three also expressed serious concerns about clarity and none could identify a concrete, sound result that the paper contributes to the field.